# Mouse models of Japanese encephalitis virus infection: A systematic review and meta-analysis using a meta-regression approach

Tehmina Bharucha[1,2‡], Ben Cleary[3‡], Alice Farmiloe[3], Elizabeth Sutton[3], Hanifah Hayati[4], Peggy Kirkwood[3], Layal Al Hamed[3], Nadja van Ginneken[5], Krishanthi S. Subramaniam[3], Nicole Zitzmann[1], Gerry Davies[3,6], Lance Turtle[3,6]*

**1** Oxford Glycobiology Institute, Department of Biochemistry, University of Oxford, Oxford, United Kingdom, **2** Lao-Oxford-Mahosot Hospital-Wellcome Trust Research Unit (LOMWRU), Microbiology Laboratory, Mahosot Hospital, Vientiane, Lao PDR, **3** Institute of Infection, Veterinary and Ecological Sciences, Faculty of Health and Life Sciences, University of Liverpool, Liverpool, United Kingdom, **4** Department of Neurology, Faculty of Medicine, Gadjah Mada University, Dr Sardjito Hospital, Yogyakarta, Indonesia, **5** Department of Primary Care and Mental Health, University of Liverpool, Liverpool, United Kingdom, **6** Tropical and Infectious Disease Unit, Liverpool University Hospitals NHS Foundation Trust (member of Liverpool Health Partners), Liverpool, United Kingdom

‡ These authors share first authorship on this work.
* lance.turtle@liverpool.ac.uk

**Data Availability Statement:** Data are available as supplemental information.

## Abstract

### Background

Japanese encephalitis (JE) virus (JEV) remains a leading cause of neurological infection across Asia. The high lethality of disease and absence of effective therapies mean that standardised animal models will be crucial in developing therapeutics. However, published mouse models are heterogeneous. We performed a systematic review, meta-analysis and meta-regression of published JEV mouse experiments to investigate the variation in model parameters, assess homogeneity and test the relationship of key variables against mortality.

### Methodology/ Principal findings

A PubMed search was performed up to August 2020. 1991 publications were identified, of which 127 met inclusion criteria, with data for 5026 individual mice across 487 experimental groups. Quality assessment was performed using a modified CAMARADES criteria and demonstrated incomplete reporting with a median quality score of 10/17. The pooled estimate of mortality in mice after JEV challenge was 64.7% (95% confidence interval 60.9 to 68.3) with substantial heterogeneity between experimental groups (I^2 70.1%, df 486). Using meta-regression to identify key moderators, a refined dataset was used to model outcome dependent on five variables: mouse age, mouse strain, virus strain, virus dose (in $\log_{10}$PFU) and route of inoculation. The final model reduced the heterogeneity substantially (I^2 38.9, df 265), explaining 54% of the variability.

**Funding:** TB is supported by the University of Oxford and the Medical Research Council [grant number MR/N013468/1]. LT is a Wellcome clinical career development fellow, supported by grant number 205228/Z/16/Z, and the NIHR Health Protection Research Unit in emerging and zoonotic infections (NIHR200907) at University of Liverpool in partnership with Public Health England (PHE), in collaboration with Liverpool School of Tropical Medicine and the University of Oxford. LT is based at University of Liverpool. The views expressed are those of the author(s) and not necessarily those of the NHS, the NIHR, the Department of Health or Public Health England. NZ is supported by the Oxford Glycobiology endowment. The funders had no role in study design, data collection and analysis, decision to publish, or preparation of the manuscript.

**Competing interests:** The authors have declared that no competing interests exist.

## Conclusion/ Significance

This is the first systematic review of mouse models of JEV infection. Better adherence to CAMARADES guidelines may reduce bias and variability of reporting. In particular, sample size calculations were notably absent. We report that mouse age, mouse strain, virus strain, virus dose and route of inoculation account for much, though not all, of the variation in mortality. This dataset is available for researchers to access and use as a guideline for JEV mouse experiments.

## Author summary

Japanese encephalitis (JE) virus (JEV) remains a leading cause of brain infection across Asia, resulting in considerable death and disability. No effective treatment exists. Mouse models are fundamental to evaluate novel treatments. We aimed to perform the first systematic literature review and data synthesis of JEV infection in mouse models. We identified an abundance of experimental data in the field, with 127 studies meeting the inclusion criteria involving a total of 5026 individual mice. Overall, 64.7% of mice died after JEV infection. However, there was incomplete reporting in publications and considerable variability in the results. In summary, the findings support the ongoing use of mouse models of JEV infection and inform researchers in the field in refining their experiments. Key factors affecting variation in mortality across studies that need to be carefully considered in study design are mouse age, mouse strain, virus strain, virus dose and route of inoculation. We highlight the need for researchers to adhere to reporting guidelines in preparing manuscripts for publication.

## Introduction

Japanese encephalitis (JE) virus (JEV) remains a leading cause of neurological infection across Asia [1,2]. The single stranded, positive sense RNA virus is a member of the genus *Flavivirus*, and groups into five genotypes (I-V) based on genomic sequence, with human infections primarily caused by genotypes I and III [3]. JEV transmission occurs in enzootic cycles between mosquitoes, pigs and birds, with human infection arising primarily from infected *Culex* spp. mosquitoes [4]. There are an estimated 100,000 cases of JEV infection per annum [5], with recent dynamic modelling suggesting that in 2021 there will be 23,600 deaths and a loss of 2,500,000 disability adjusted life years [5,6]. Moreover, JE predominantly affects children in endemic areas and has devastating socioeconomic consequences.

A handful of randomised clinical trials of treatments for JE have been performed to date, however no effective treatment has been identified [7]. Multiple vaccines exist and are recommended by the World Health Organisation (WHO) [8]. Although recent efforts have strengthened JE vaccination programs, still only 12 of 24 endemic countries include JE vaccine in routine immunisation policies; even then, it is not uniformly nationwide, with vaccine coverage in targeted areas reported to be as low as 39% [1]. As viremia in JE is too low to propagate, humans are considered dead-end hosts and the infection is zoonotic, which provides additional challenges as vaccination alone is therefore not sufficient for JE eradication [4].

The high lethality of JE and absence of effective therapies mean that animal models are crucial in developing our understanding of the disease and therapeutic prospects. There have

been reports published of experiments using JEV in mouse models dating back to 1935 when the virus was first isolated [9,10]. Mice are frequently the preferred model for studying human infections due to their low-cost, timely reproduction and variability [11]. In the last decade, many studies of JEV have used animal models to address a wide variety of different questions, such as the role of various components of the immune system in protection from JEV, pathogenesis of JE and for testing treatments. These studies have contributed greatly towards further understanding JE pathogenesis [7]. However, mouse models of JE have not been standardised and can be highly variable across laboratories contributing to contradictory results [12].

No systematic review of mouse models of JE has been conducted. Therefore, in order to understand which model parameters account for variation, and to ensure homogeneous reporting of results, we conducted a systematic review of published experiments using JEV in mouse models. We hypothesised that virus strain, virus dose, route of administration, mouse strain, mouse age and mouse sex would influence lethality. We therefore aimed to test the relationship of these variables on mortality from JEV infection in mice and to develop guidance on the set up and reporting of mouse models of JE.

## Methods

The study adhered to PRISMA guidelines for systematic reviews and the protocol is included in supplementary data (S1 Data Protocol). A PubMed search was performed using the terms ("encephalitis, japanese"[MeSH Terms] OR ("encephalitis"[All Fields] AND "japanese"[All Fields]) OR "japanese encephalitis"[All Fields] OR ("japanese"[All Fields] AND "encephalitis"[All Fields]) OR "je"[All Fields] OR "jev"[All Fields]) AND ("mice"[MeSH Terms] OR "mice"[All Fields] OR "mouse"[All Fields] OR "mice"[MeSH Terms] OR "mice"[All Fields] OR "mus"[All Fields]). The search date ranged from 1935 (the year of first isolation of JEV) to August 2020 and only English language text was included. Retrieved references were downloaded into EndNote for the removal of duplicate studies and the abstracts and/or full-text screened by two authors independently for eligibility as per the criteria detailed below, see Fig 1 for the PRISMA flow diagram.

The inclusion criteria were any publication that included an experiment meeting the following criteria: 1. JEV was inoculated into mice; 2. virus dose was reported; 3. JEV strain or source was reported; 4. immunocompetent mice were used and the strain was reported; 5. mortality was reported as either death or humane endpoint (primary analysis & outcome measures, secondary outcome measures 1 and 2) or other pathological outcome reported (secondary outcome 3); 6. published in English; and 7. primary research. Additional data were also extracted (see S1 Data Protocol) but did not serve as an exclusion from the primary analysis. Studies were excluded if they reported inoculation using non-pathogenic JEV (for example the vaccine strain SA14-14-2) only, or if data on individual animals was not reported and it was not possible to extract the data. In order to minimise non-specific immune effects, data were collected only on groups of mice that received JEV only, and no other material. For example, data were frequently derived from reports that tested a treatment or vaccine, in these cases only the control group was extracted.

Quality assessment and data extraction were performed by two authors independently, using standardised excel sheet proformas. The quality of studies was assessed based on ten standard quality measures used previously for animal model meta-analyses by CAMARADES (the Collaborative Approach to Meta-Analysis and Review of Animal Data in Experimental Studies) [12]. The CAMARADES checklist is used to perform a combined assessment of the reporting of a number of measures to reduce bias, and several indicators of external validity and study quality. Additional data extracted as study quality measures were: statement of the

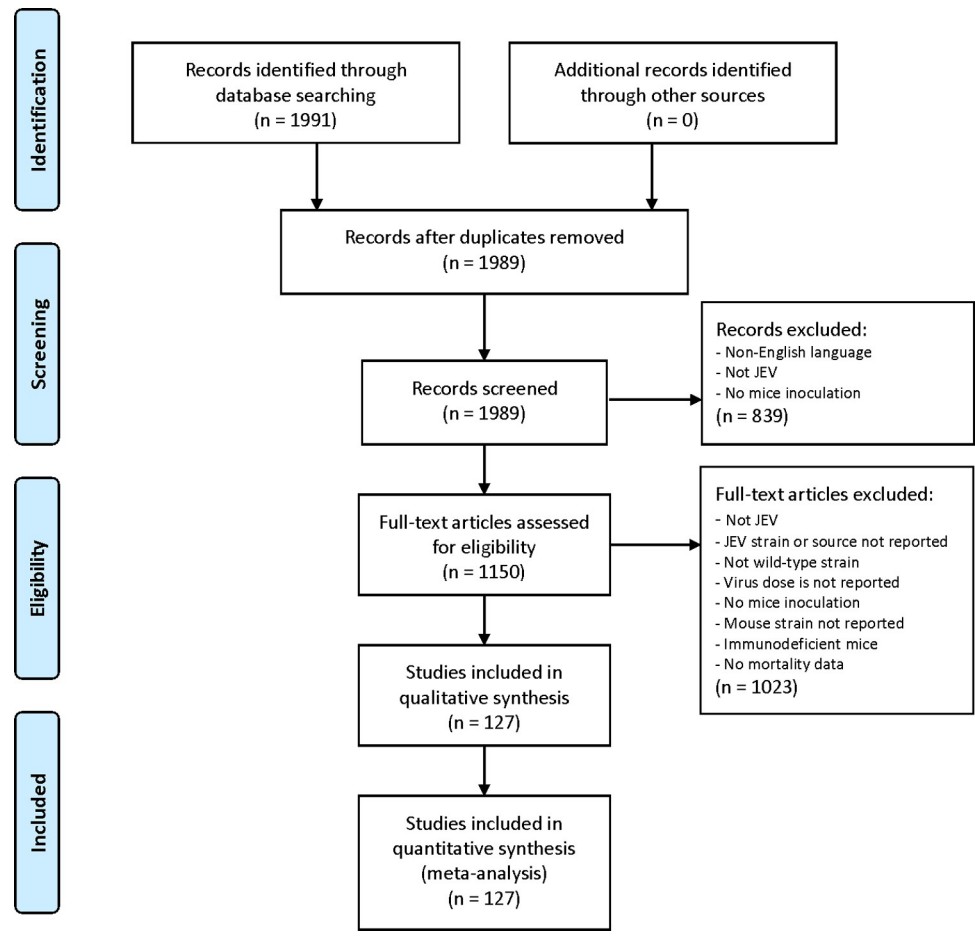

**Fig 1. PRISMA flow diagram of study selection.** Preferred Reporting Items for Systematic Reviews and Meta-Analyses (PRISMA; http://www.prisma-statement.org/) flow diagram showing the stages of study identification and selection. 127 studies met the eligibility criteria and were included in the meta-analysis. For each quality measure, fulfilling the criteria was interpreted as a score of 1 and 'low risk of bias'.

intent of the experiments conducted, mouse strain used, virus used, dose of virus given, route of inoculation given, age of mice at inoculation explicitly stated or easily calculated (e.g., "mice were immunised at 6 weeks of age . . . and challenged 2 weeks after immunisation") and cell type/tissue that the virus was derived from. For each quality measure, fulfilling the criteria was interpreted as a score of 1 and 'low risk of bias'. Each study was allocated a score out of 17 based on these quality measures. Sensitivity analysis was performed on the final model using subgroups with different overall quality scores.

Data were analysed using R version 4.0.2 [13]. Variables of interest were plotted against mortality at the level of individual experiments. Publication bias was explored using funnel plots. Individual and aggregated forest plots were used to summarise data, stratified by key variables. Meta-regression was used to quantify the impact of experiment-level covariates on heterogeneity of outcomes. The generalised $R^2$ (explained variance) and $I^2$ (total heterogeneity/variability) statistics, likelihood ratio test, and Akaike's Information Criterion (AIC) were used to judge model fit. Routine regression diagnostics were used to test model assumptions. A multivariable analysis was performed using all of the variables in a single model to estimate mortality. Sensitivity analyses were used to consider the strength of effect of each variable

individually using a forward stepwise approach, i.e., assessing the iterative effect of including them in the model.

## Results

### Summary

The initial search identified 1991 articles, of which 127 were included in the review, detailed in the PRISMA flow diagram (Fig 1). Data were available on 5026 individual mice in 487 experimental groups challenged with JEV. Studies involved a median of two (IQR 1–4; range 1–84) groups where JEV was inoculated into mice with no other treatment or substance; one study performed by Miura *et al.* in 1988 [14] represented an outlier with 84 groups involving a total of 527 mice. Experimental groups included a median of 10 (IQR 5–12; range 2–112) mice; 18 (3.7%) with over 20 mice, and two (0.4%) with over 50 mice. No study performed in the last 5 years included an experimental group with over 20 mice. Between 1970 and 2020 studies were conducted in 12 countries, primarily in Asia (China, India, Japan, Republic of Korea, Singapore, Malaysia, Taiwan), but also in Australia, Europe (U.K., France, The Netherlands) and N. America (U.S.A.) (S2 and S3 Figs). Eighty-four last authors were identified in 54 institutions, with almost half (25; 46%) of institutions publishing multiple studies.

Compared with CAMARADES criteria, several pieces of information in specific areas were not reported. Less than 50% of studies included statements that allocation to experimental group was random, treatment and outcome were blinded to investigator, neuroprotective anaesthesia was used and no conflict of interests were present. No study reported having performed a sample size calculation or that the temperature was controlled during experiments. Median quality score was 10 (IQR 9–11, range 7–13) across the 17 criteria (Fig 2). Data extracted are displayed in Table 1, with details of the variables and missing study-level characteristics.

The global estimate of mortality in mice after JEV challenge, i.e., the base meta-regression model, was 64.7% (95% confidence interval 60.9 to 68.3). There was substantial heterogeneity (I^2 70.1%, df 486) between experimental groups. Therefore, we next determined the influence of key variables to develop a final meta-regression model.

### Analysis of the moderating effect and interactions of key variables

**Mouse strain.** Fourteen different mouse strains were used in the studies, see Fig 3A. Six strains were used for the vast majority (4771/5026; 95%) of studies; C57BL/6, BALB/c, Swiss, C3H/He, ICR and ddY mice. There were less than 100 mice (range 6–65) studied for each of the eight remaining strains. The mortality of JEV challenge in different mouse strains was highly variable (Fig 3B). Overall, BALB/c, Swiss and ICR were more susceptible to JEV than other mouse strains and ddY was more resistant. Incorporating mouse strain as a moderator in the base meta-regression model reduced the variability of the base model (I^2 63.6%, S1 Data). Analysis of the top six mouse strains (S2 Data) suggested interactions between the variables, for example mortality in ICR mice is higher than C57BL/6; however, the median age of the mice used in experiments with the different strains is different at 14 (IQR 3–21) vs 42 (IQR 35–95) days respectively.

**Mouse age.** Mouse age was categorised into months. Mice used in JEV challenge studies were predominantly younger (3754/5026; 75%), i.e., less than 3 months old (Fig 3C). Mortality declined with the age of the mice (Fig 3D). Incorporating mouse age as a moderator in the base meta-regression model reduced the variability of the base model (I^2 65.2%, S1 Data).

**Mouse sex.** Sixty-eight studies (54%) reported the mouse sex, data were missing for 2686/5026 mice (53%). The vast majority of mice for which data were available were female (2299;

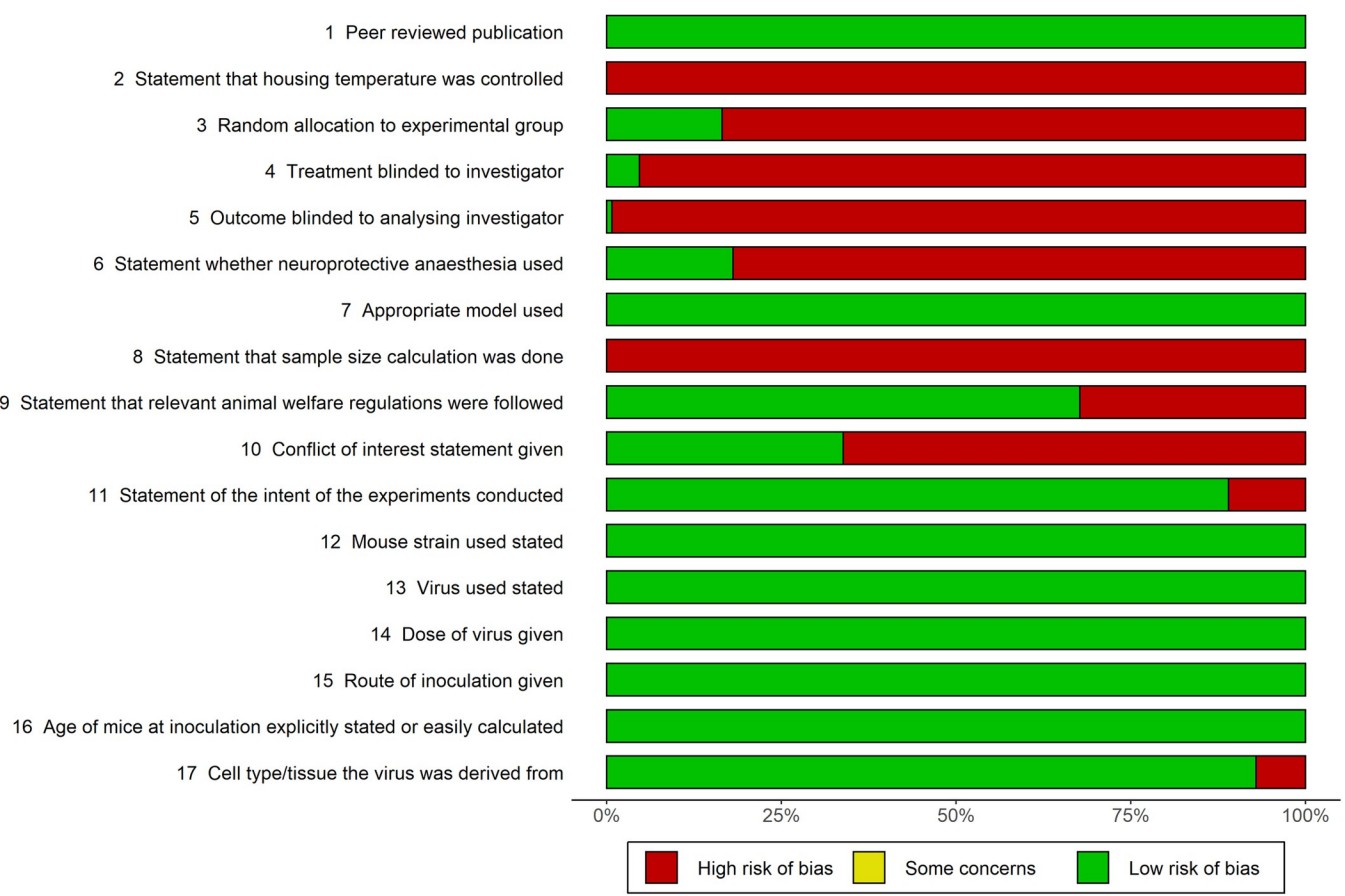

**Fig 2. Quality assessment of included studies.** The quality of included studies was assessed using a 17-point modified CAMARADES checklist ([https://www.equator-network.org/reporting-guidelines/collaborative-approach-to-meta-analysis-and-review-of-animal-data-from-experimental-studies-camarades/](https://www.equator-network.org/reporting-guidelines/collaborative-approach-to-meta-analysis-and-review-of-animal-data-from-experimental-studies-camarades/)).

**Table 1. Variables with data extracted.**

| Variable | Details | No. (%) of studies with missing data, i.e., no reporting |
|---|---|---|
| Year of publication | 6 categories | 0 (0%) |
| Last author | 84 categories | 0 (0%) |
| Institution | 54 categories | 0 (0%) |
| Country | 12 categories | 0 (0%) |
| Mouse strain* | 14 categories | 0 (0%) |
| Mouse age | 4 categories | 2 (1.6%) |
| Mouse sex | 2 categories | 59 (56%) |
| Virus genotype | 3 categories | 0 (0%) |
| Virus strain* | 38 categories | 0 (0%) |
| Virus dose in PFU* | $\log_{10}$PFU continuous | 29 (22.8%) |
| Route of administration | 8 categories | 0 (0%) |

*The variable was a criteria for inclusion in the systematic review. For the purpose of meta-regression analysis, doses in tissue culture infective dose ($TCID_{50}$) were converted to plaque forming units (PFU) and doses in lethal dose (LD) were excluded.

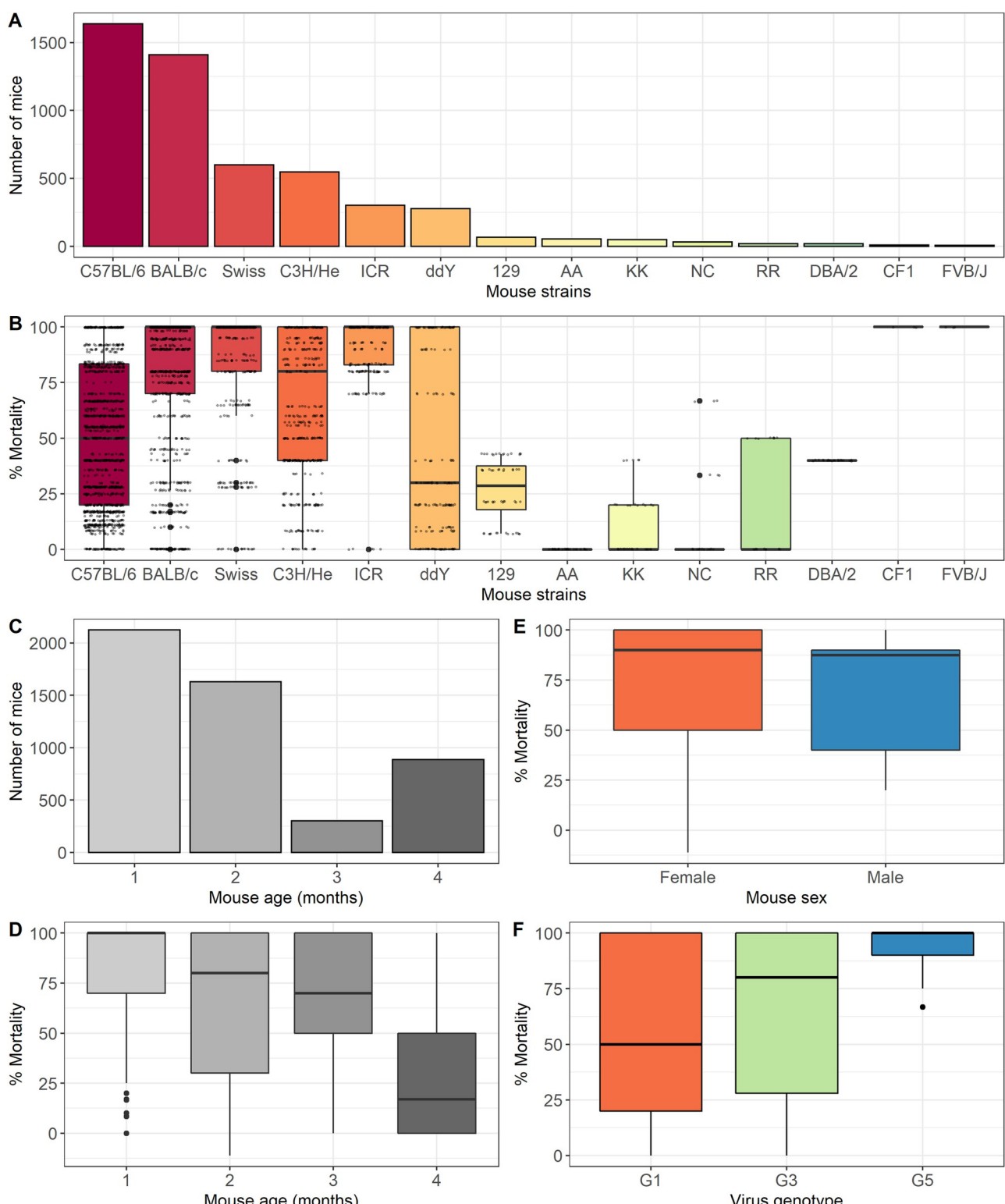

**Fig 3. Summary of the number of mice of different mouse strains used.** A: Bar chart of the number of mice of different strains included in the meta-analysis (n = 5026). B: Box plot of the percentage mortality from JEV challenge in different strains of mice included in the meta-analysis (n = 5026). C: Bar chart of the number of mice of different ages in months included in the meta-analysis (n = 5026). D: Box plot of the percentage mortality from JEV challenge in different ages in months of mice included in the meta-analysis (n = 5026). E: Box plot of the percentage mortality from JEV challenge in male vs. female mice included in the meta-analysis (n = 5026). F: Box plot of the percentage mortality of mice included in the meta-analysis (n = 5026) from challenge with different JEV genotypes.

98%). Mortality in male vs. female mice following JEV challenge was not significantly different (Fig 3E). Incorporating mouse sex as a moderator in the base meta-regression model minimally reduced the variability of the base model (I^2 68.7%, S1 Data).

**Virus genotype.**   Three of the five known JEV genotypes were used in studies; predominantly genotype 3 (4407; 88%), to a lesser extent genotype 1 (515; 10%) and rarely genotype 5 (104; 2%). Mortality of mice challenged with different genotypes is shown in Fig 3F. Incorporating virus genotype as a moderator in the base meta-regression model minimally reduced the variability of the base model (I^2 69.9%), see S1 Data.

**Virus strain.**   Thirty-eight different JEV strains were used in the studies (Fig 4A); 7 (18%) genotype 1, 29 (77%) genotype 3 and 2 (5%) genotype 5. Twenty of 127 studies (16%) reported accession numbers but none reported sequencing data to confirm the virus used. There were 16 strains with identifiable accession numbers, 12 full and 4 partial sequences; there did not appear to be any particular genetic selection bias given the relationship to other virus isolates on a phylogenetic tree (S1 Fig).

Ten JEV strains were used in less than 100 mice each, such that 13 JEV strains represented 4004/5026 (80%) of the mice included in the review. The mortality of JEV challenge with

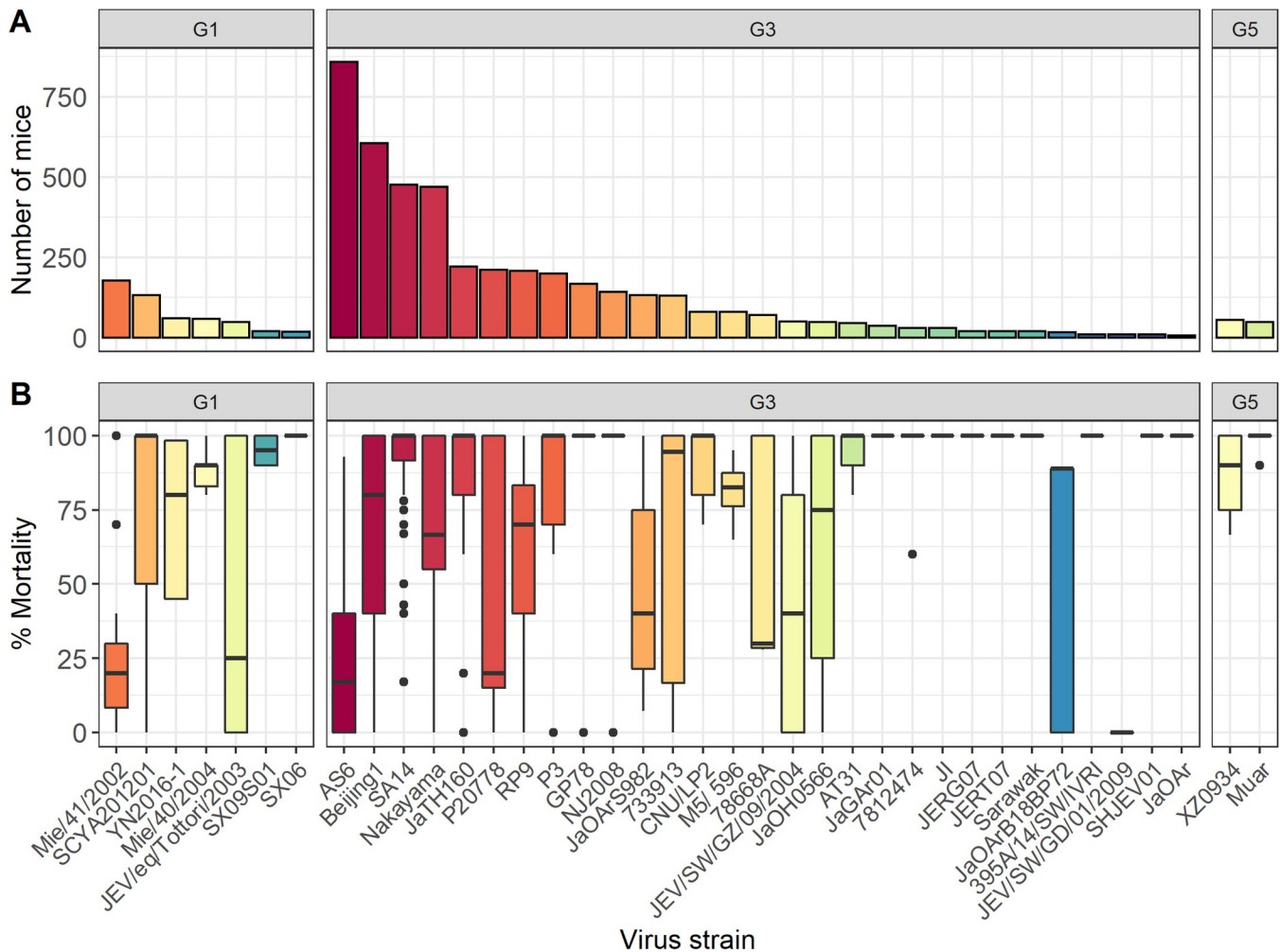

**Fig 4. Summary of the number of mice that received different JEV strains.** A: Number of mice included in the meta-analysis (n = 5026) that received different JEV strains grouped by virus genotype. B: Box plot of the percentage mortality of mice included in the meta-analysis (n = 5026) from challenge with different JEV strains grouped by virus genotype.

different virus strains was highly variable (Fig 4B). Incorporating virus strain as a moderator in the base meta-regression model reduced the variability of the base model (I^2 56.5%, S1 Data). Analysis of the top 13 JEV strains (S3 Data) suggested interaction between the variables, for example mice challenged with AS6 have significantly lower mortality than those challenged with Beijing1, however the median dose of AS6 is over 6000 times lower than that of Beijing1.

**Virus dose.**   Ninety-three (73%) studies reported the dose in plaque forming units (PFU), 28 (22%) in lethal dose (LD), 5 (4%) in median tissue culture infectious dose ($TCID_{50}$) and 1 (1%) in median cell culture infectious dose ($CCID_{50}$). $TCID_{50}$ was converted to PFU (PFU = 0.7 x $TCID_{50}$) and the analysis was performed with the virus dose in $log_{10}$PFU, using this approach there were missing data for 1239/5026 (25%) mice included in the review. There was a normal distribution of doses used in the studies (Fig 5A). There was no clear relationship between dose and mortality, however performing a subgroup analysis by route did demonstrate an association, see Figs 5B and 6. Incorporating virus dose in PFU as a moderator in the base meta-regression model reduced the variability of the base model (I^2 65.9%), see S1 Data.

**Route of administration.**   Eight routes of JEV challenge were used across the included studies; intracerebral (IC), sub-cutaneous (SC), intranasal (IN), conjunctival (CONJ), intravenous (IV), intramuscular (IM), intraperitoneal (IP) and IP with sham IC, see Fig 7A. IN and CONJ were used in only 64 (1.3%) and 10 (0.2%) mice respectively. There was no significant difference in mortality in mice challenged by different routes of administration (Fig 7B). Incorporating route of administration as a moderator in the base meta-regression model minimally reduced the variability of the base model (I^2 68.7%), see S1 Data. Analysis of the top 6 routes of administration (S4 Data) suggests interaction between these variables.

## Interactions

Analysis was performed to examine interactions of key variables. There were no clearly identifiable interactions. This was in part due to a lack of data for all the categories; for example, as seen in Fig 8, investigators use mice of the same age for specific mouse strains.

## Meta-regression analysis

In order to reduce instability in the model due to missing study-level characteristics and to better explore interactions between the variables, a sub-group analysis was performed using the top six mouse strains, top 13 virus strains, and top six routes of inoculation. The reduced dataset included 73 (57.5%) of the studies, 265 (54.4%) of the experimental groups and 2770/5026 (55.1%) of the mice in the full dataset. The pooled estimate of mortality in mice after JEV challenge, i.e., the base meta-regression model for the reduced dataset was 64.2% (95% confidence interval 59.2 to 68.9). There was substantial heterogeneity (I^2 70.0%, df 264) between experimental groups. In view of the analysis results of the key variables and biological plausibility of their role in moderating the effect, the final model used the reduced dataset. This reduced the heterogeneity substantially (I^2 38.9%, df 265), see Table 2 and the forest plot in Fig 9. This confirmed our starting hypothesis that mouse strain, mouse age, route of administration, virus dose and virus strain account for much of the variation in mortality in mouse models of JE, whereas there were not sufficient data to draw conclusions about mouse sex. A sensitivity analysis using subgroups with different thresholds for overall quality scores was performed and did not improve the heterogeneity.

## Discussion

The results highlight the wealth of data available on mouse models of JE, with 127 studies meeting the eligibility criteria for inclusion, totalling 5026 mice in 487 experimental groups.

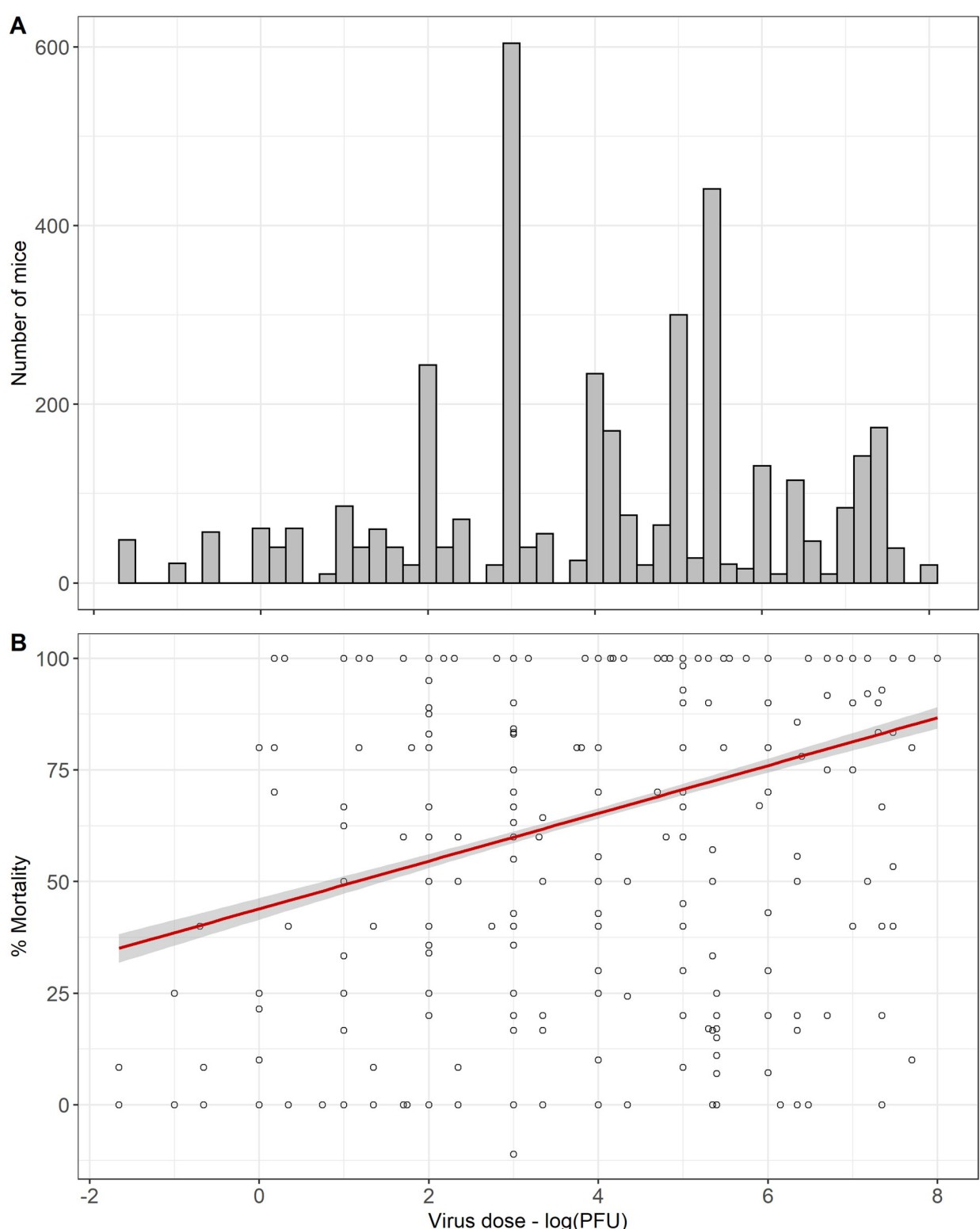

**Fig 5. Summary of the number of mice that received different doses.** A: Histogram of the number of mice included in the meta-analysis (n = 5026) that received different JEV doses in plaque forming units (PFU). B: Scatter plot of the percentage mortality of mice included in the meta-analysis (n = 5026) that received different JEV doses in plaque forming units (PFU).

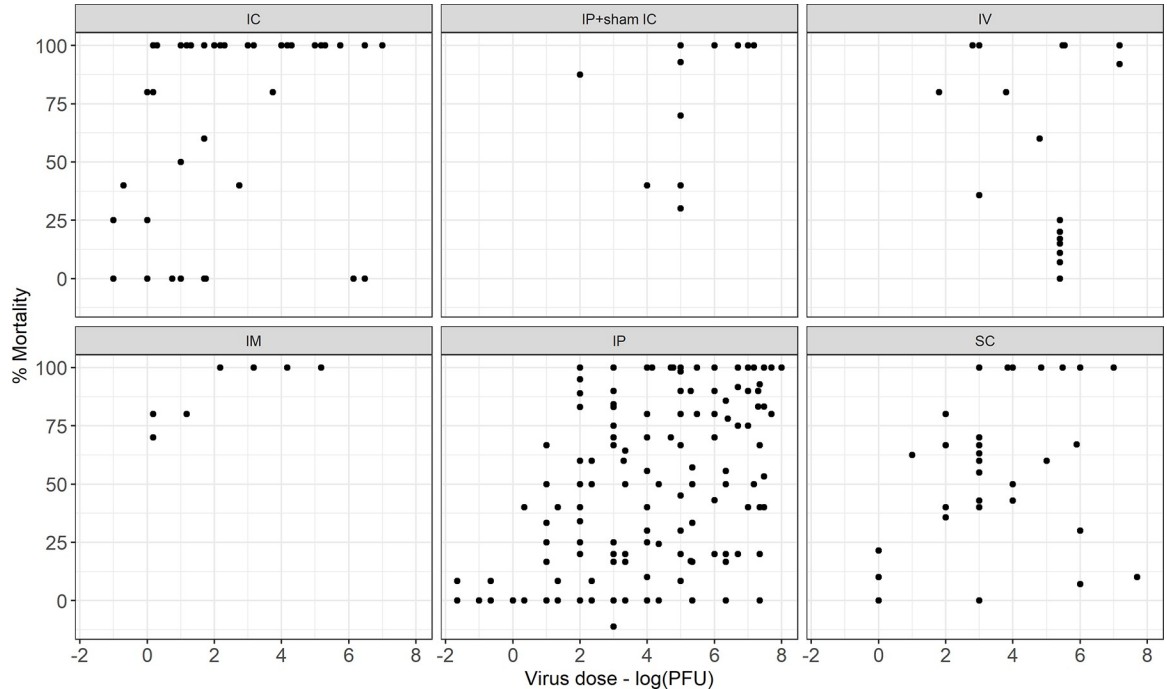

**Fig 6. Mortality in mice following JEV challenge against dose in log$_{10}$PFU grouped by the route of inoculation.** Scatter plot of percentage mortality of mice included in the meta-analysis (n = 5026) that received different JEV doses in plaque forming units (PFU), grouped by the route of administration.

The pooled estimate for mortality was 64.7% (95% confidence interval 60.9 to 68.3%). Mouse strain, mouse age, route of administration, virus dose and virus strain account for much of the variation in mortality in mouse models of JE. The large number of mice studied to date, the significant unresolved heterogeneity across studies and the ongoing knowledge gaps in our understanding of JE (particularly the lack of effective treatment) underlines the importance of this systematic review to inform future research. In particular, the analysis aims to refine future mouse models of JEV infection, to improve the quality of future studies and reduce unnecessary replication. The extracted data has been made publicly available to enable researchers to perform their own analysis (S5 Data).

Quality assessment was relatively low with a median quality score of 10 (IQR 9–11, range 7–13) across the 17 criteria. However, there was an improvement in the score over time. This highlights the need to reduce risk of bias through detailed reporting, following CAMARADES guidelines. In particular, there is a need for performance and reporting of blinding, randomisation and *a priori* sample size calculations. None of the studies were excluded based on these quality assessments. Beyond the quality assessment criteria, analysis of data extracted demonstrated missing data for study-level characteristics, most strikingly for reporting of mouse sex in 59/127 (56%) studies. Furthermore, accession numbers were only reported in 20/127 (16%) studies with no study reporting sequencing of the actual JEV strain used, which is important given the potential for sequence variability after serial passage in culture.

The number of mice per experimental group has become more consistent with time, as researchers are more aware and institutional guidance mandates reductions in experimental group sizes. Nonetheless, it is always a balance between sufficient power versus minimising numbers used/ sacrificed. In mouse experiments, it is typical to use 5 mice for inbred strains

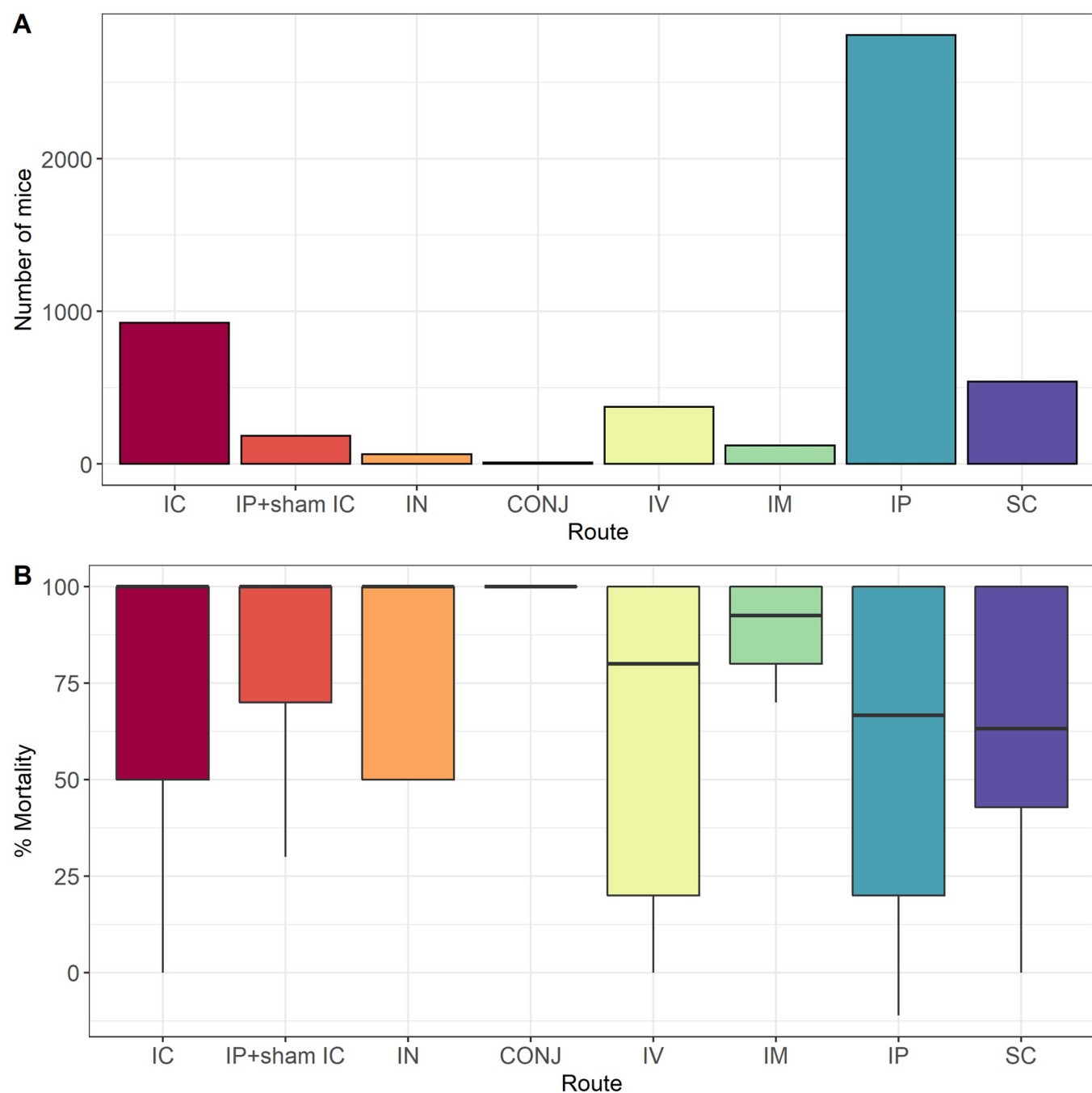

**Fig 7. Summary of the number of mice and mortality in mice challenged by different routes of administration.** A: Bar chart of the number of mice included in the meta-analysis (n = 5026) that were inoculated with JEV by different routes. B: Box plot of the percentage mortality of mice included in the meta-analysis (n = 5026) that were inoculated with JEV by different routes.

and 8–10 for the outbred, however based on our findings, a power calculation suggests that larger numbers are needed. For example, to detect a halving of mortality from an intervention with an overall mortality of 64.7%, 35 mice are needed per group to give 80% power at the 5% significance level.

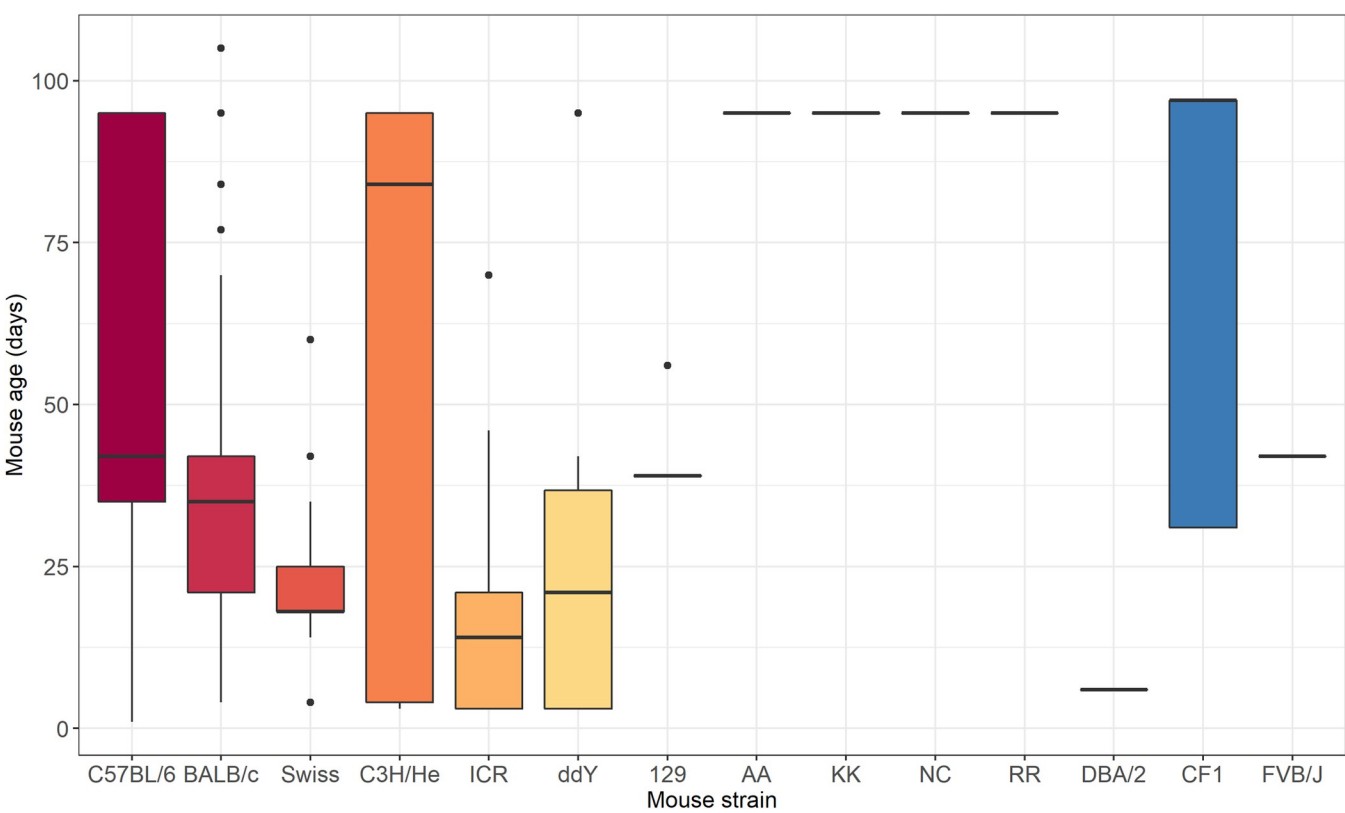

**Fig 8. Box plot of the age distribution (in days) of mice across mice strains used in the included studies (n = 5026).**

### Analysis of moderating effect of key variables

**Mouse strain.**   Unsurprisingly, C57BL/6 (black inbred strain; 1637/5026–33%) and BALB/c (white inbred strain; 1409/5026–28%) were the most common strains used; these are well-characterised, reproducible, easily available and cheap [15]. C57BL/6 mice have a tendency to generate Th-1 immune responses, whereas BALB/c are skewed towards Th-2 responses. Furthermore, the data were consistent with existing dogma that BALB/c are more susceptible to infection than C57BL/6, with median survival (IQR) 0% (0–30) and 50% (17–80) respectively. Although the analysis was restricted to six strains with more than 100 mice in each group (C57BL/6, BALB/c, Swiss, C3H/He, ICR and ddY mice; 4771/5026–95%), all were susceptible to JEV infection. It was not possible to resolve individual mouse strains, as there is significant variation even between sub-strains based on breeding history, original parent strains, and source locations.

**Mouse age.**   JEV predominantly affects children in endemic areas, which most likely reflects early life exposure leading to immunity by adulthood [16], though adults are also susceptible to JE upon first exposure [17]. Additional factors that are likely to play a role include the development of the immune system as well as maturation of blood brain barrier; to what extent this is relevant to human disease is not clear, but in mice this is clearly relevant, with some investigators using a sham IC injection in order to disrupt the blood brain barrier and allow encephalitis to develop [18]. The systematic review provides support for this, as mortality reduced with age.

**Virus strain.**   Variation of the virus is a plausible explanation for variation between models. Genotype 3 (G3) JEV remains the most commonly isolated JEV genotype from human

**Table 2. Final meta-regression model with filtered dataset consisting of the top six mouse strains, top 13 virus strains, virus dose in PFU and top six routes of inoculation.** This included 73/127 (57.5%) of the studies, 265/487 (54.4%) of the experimental groups and 2770/5026 (55.1%) of the mice in the full dataset.

| Mixed-Effects Model (k = 265; tau^2 estimator: DL) | | | | |
|---|---|---|---|---|
| logLik | deviance | AIC | BIC | AICc |
| -452.533 | 376.5373 | 961.0668 | 1061.299 | 967.9481 |
| tau^2 (estimated amount of residual heterogeneity) | | | 0.5520 (SE = 0.1399) | |
| tau (square root of estimated tau^2 value) | | | 0.7430 | |
| I^2 (residual heterogeneity / unaccounted variability) | | | 38.92% | |
| H^2 (unaccounted variability / sampling variability) | | | 1.64 | |
| Test for Residual Heterogeneity | | | QE (df = 238) 389.6412, p-val < .0001 | |
| Test of Moderators (coefficients 1:27) | | | QM (df = 27) 365.4701, p-val < .0001 | |

|  | estimate | se | zval | pval | ci.lb | ci.ub | |
|---|---|---|---|---|---|---|---|
| routeIP+sham IC | -2.6283 | 1.1402 | -2.3051 | 0.0212 | -4.8631 | -0.3936 | * |
| routeIV | -2.469 | 1.1408 | -2.1642 | 0.0304 | -4.705 | -0.233 | * |
| routeIM | -1.5765 | 1.2807 | -1.2309 | 0.2184 | -4.0866 | 0.9337 | |
| routeIP | -2.4263 | 1.0517 | -2.307 | 0.0211 | -4.4876 | -0.365 | * |
| routeSC | -2.8807 | 1.0965 | -2.6273 | 0.0086 | -5.0298 | -0.7317 | ** |
| mouse_strain_2BALB/c | 0.1643 | 0.3244 | 0.5063 | 0.6126 | -0.4716 | 0.8001 | |
| mouse_strain_2Swiss | 1.9913 | 0.5367 | 3.7104 | 0.0002 | 0.9394 | 3.0432 | *** |
| mouse_strain_2C3H/He | 1.5426 | 0.3646 | 4.2312 | < .0001 | 0.828 | 2.2572 | *** |
| mouse_strain_2ICR | 0.9151 | 0.5831 | 1.5693 | 0.1166 | -0.2278 | 2.058 | |
| mouse_strain_2ddY | 0.4711 | 0.4221 | 1.1163 | 0.2643 | -0.3561 | 1.2984 | |
| mouse_age_mths2 | -0.6473 | 0.279 | -2.3196 | 0.0204 | -1.1942 | -0.1004 | * |
| mouse_age_mths3 | -0.353 | 0.4045 | -0.8727 | 0.3828 | -1.1458 | 0.4398 | |
| mouse_age_mths4 | -0.3302 | 1.0482 | -0.315 | 0.7528 | -2.3845 | 1.7242 | |
| logPFU | 0.2425 | 0.0476 | 5.0958 | < .0001 | 0.1492 | 0.3357 | *** |
| virus_strain_resolvedBeijing1 | 1.771 | 1.0115 | 1.7509 | 0.08 | -0.2114 | 3.7535 | . |
| virus_strain_resolvedSA14 | 2.3219 | 1.1145 | 2.0833 | 0.0372 | 0.1374 | 4.5063 | * |
| virus_strain_resolvedNakayama | 3.0299 | 1.0707 | 2.83 | 0.0047 | 0.9315 | 5.1284 | ** |
| virus_strain_resolvedGP78 | 4.1023 | 1.1999 | 3.419 | 0.0006 | 1.7506 | 6.454 | *** |
| virus_strain_resolvedJaTH160 | 3.0549 | 1.1134 | 2.7437 | 0.0061 | 0.8726 | 5.2372 | ** |
| virus_strain_resolvedP20778 | 0.6102 | 1.1323 | 0.5389 | 0.5899 | -1.6091 | 2.8295 | |
| virus_strain_resolvedRP9 | 2.001 | 1.0649 | 1.8791 | 0.0602 | -0.0861 | 4.0882 | . |
| virus_strain_resolvedP3 | 2.3007 | 1.1275 | 2.0406 | 0.0413 | 0.0909 | 4.5105 | * |
| virus_strain_resolvedMie/41/2002 | -0.5313 | 1.1207 | -0.4741 | 0.6354 | -2.7277 | 1.6652 | |
| virus_strain_resolvedNJ2008 | 4.0609 | 1.4212 | 2.8573 | 0.0043 | 1.2754 | 6.8464 | ** |
| virus_strain_resolvedJaOArS982 | 2.1473 | 1.1308 | 1.8989 | 0.0576 | -0.069 | 4.3636 | . |
| virus_strain_resolvedSCYA201201 | 0.9755 | 1.2492 | 0.781 | 0.4348 | -1.4728 | 3.4238 | |

Signif. codes: 0 '***' 0.001 '**' 0.01 '*' 0.05 '.' 0.1 ' ' 1

cases; accepting that virus isolation from cases is a rare event and is also the most frequently used in mouse models of JEV infection [19]. Many different strains of JEV have also been used in mouse models. These are rarely sequenced by the investigators, meaning it was not possible to compare sequences and/or resolve the strains used. The top 13 virus strains, relatively well-characterised sequences, constituted 80% of all mice included in the systematic review, and the viral strain accounted for 14% of the variability of the models.

**Virus dose.** Although the ideal reporting of virus dose is in quantified infectious units, or PFU, many papers do not use this measure and instead report the $TCID_{50}$ or $LD_{50}$. Genome copies measured by RT-qPCR may also serve as a measure of viral inoculum, though in

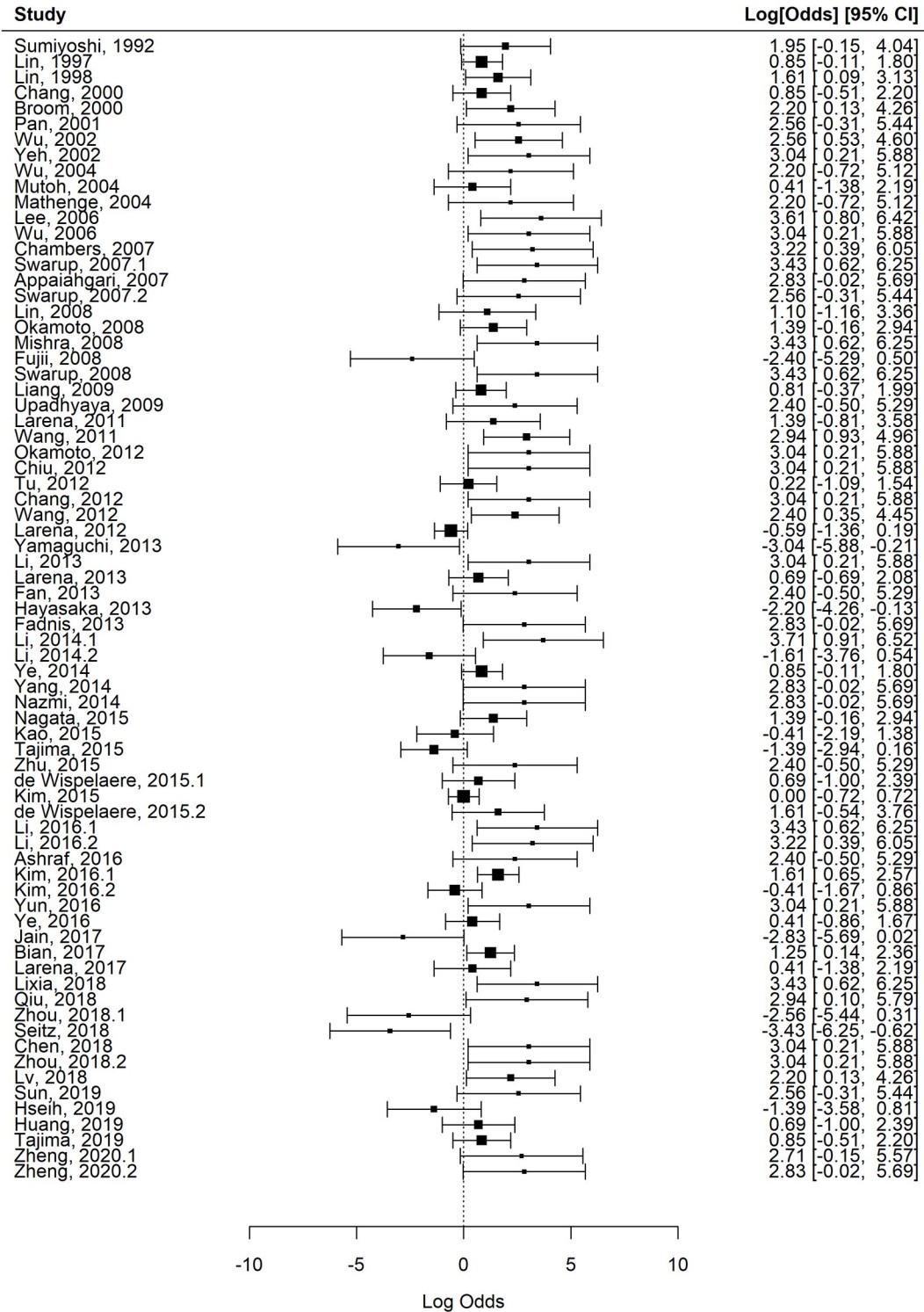

**Fig 9. Forest plot of the studies included in the final meta-regression model.** Studies are identified with the first author and year of publication. The log odds of mortality are shown along with 95% confidence intervals.

practice this is rarely used. $TCID_{50}$ is a direct measure of infectivity and can be used to approximate PFU whereas $LD_{50}$ is confounded by being influenced by the outcome variable in our analysis and cannot be approximated in this way. For this reason, there were significant missing data (1239/5026–25% mice). Nonetheless, in view of the biological plausibility of the moderating effect of dose, the dose in $log_{10}PFU$ was included in the final model.

**Route of inoculation.**   The preferred route of inoculation involves the one with the best external validity, that which is technically straightforward to perform safely with a dangerous pathogen such as JEV (which poses a risk to laboratory workers) and provides a robust and reproducible infection. The IP method was most widely used (2811/5026–56% mice), followed by IC (924/5026–18% mice) and then SC (539/5026–11%). The meta-regression analysis showed a minimal impact of the route of inoculation on reducing heterogeneity, however further exploration demonstrated that this was due to interaction of other variables. It was notable that there was an association between the proximity of the route of inoculation to the brain and mortality (p value < 0.001).

A refined dataset of 2770 mice were produced and a final meta-regression model run using mouse age, mouse strain, virus strain, virus dose in $log_{10}PFU$ and route of inoculation. The final model reduced the heterogeneity substantially (I^2 38.9%, df 265) such that 61% of the variability was explained. Despite this analysis, nearly a third of the variability in mouse models of JE remained unexplained, leaving significant room for variation due to individual laboratories, and therefore also room for improvement in standardisation of these important and useful models.

Undoubtedly there are inherent limitations in mouse models of infectious diseases that affect the external validity of the findings. Furthermore, the missing data for combinations of the different key variables reduces the internal validity. Nonetheless, this review still represents the most comprehensive body of data on mouse models of JE assembled to date. We summarise our final recommendations in Box 1. Attention to performance and reporting of experiments on key factors identified in this review will reduce heterogeneity and enable standardisation of models. This is critical to enable evaluation of novel therapeutics.

---

Box 1: Final recommendations

- Power calculations are crucial to ensure that experiments are appropriately designed to detect effects of interventions

- Factors that affect variability in outcomes and need careful attention in study design include mouse strain, mouse age, virus strain, virus dose and route.

- Virus strains used in experiments need to be sequenced and the data included in publications.

- The data has been made publicly available and serves to inform future experiments in this field.

---

## Supporting information

**S1 Data Protocol. Protocol: A systematic review of mouse models of Japanese encephalitis.** (DOCX)

**S1 Data. R code for meta-regression analysis.** (DOCX)

**S2 Data. Analysis of top 6 mouse strains.**
(XLSX)

**S3 Data. Analysis of top 13 virus strain.**
(XLSX)

**S4 Data. Analysis of top 6 routes.**
(XLSX)

**S5 Data. Study data.**
(XLSX)

**S1 Fig. JEV phylogenetic tree including all complete genome sequences uploaded to Gen-Bank (n = 329) and partial sequences (n = 4) if used in studies with sequences used in studies highlighted.** This was created using MEGA-X and genotypes assigned manually.
(PDF)

**S2 Fig. Locations (countries) of included studies (12 countries, n = 127).**
(DOCX)

**S3 Fig. Year of publication of included studies (1970–2020; n = 127).**
(DOCX)

## Acknowledgments

We thank Alan Barrett (The University of Texas Medical Branch) for useful discussion.

## Author Contributions

**Conceptualization:** Lance Turtle.

**Data curation:** Tehmina Bharucha, Ben Cleary, Alice Farmiloe, Elizabeth Sutton, Hanifah Hayati, Peggy Kirkwood, Layal Al Hamed, Lance Turtle.

**Formal analysis:** Tehmina Bharucha, Ben Cleary, Krishanthi S. Subramaniam, Nicole Zitzmann, Gerry Davies, Lance Turtle.

**Funding acquisition:** Lance Turtle.

**Investigation:** Tehmina Bharucha, Ben Cleary, Lance Turtle.

**Methodology:** Tehmina Bharucha, Ben Cleary, Nadja van Ginneken, Krishanthi S. Subramaniam, Gerry Davies, Lance Turtle.

**Project administration:** Tehmina Bharucha, Ben Cleary, Lance Turtle.

**Resources:** Lance Turtle.

**Software:** Tehmina Bharucha, Lance Turtle.

**Supervision:** Gerry Davies, Lance Turtle.

**Visualization:** Tehmina Bharucha, Ben Cleary, Gerry Davies, Lance Turtle.

**Writing – original draft:** Tehmina Bharucha, Ben Cleary, Lance Turtle.

**Writing – review & editing:** Alice Farmiloe, Elizabeth Sutton, Hanifah Hayati, Peggy Kirkwood, Layal Al Hamed, Nadja van Ginneken, Krishanthi S. Subramaniam, Nicole Zitzmann, Gerry Davies, Lance Turtle.

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
