## [Decision Letter · Decision Letter 0]

28 Sep 2021

Dear Dr Turtle,

Thank you very much for submitting your manuscript "Mouse models of Japanese encephalitis virus infection: a systematic review and meta-analysis using a meta-regression approach" for consideration at PLOS Neglected Tropical Diseases. As with all papers reviewed by the journal, your manuscript was reviewed by members of the editorial board and by several independent reviewers. The reviewers appreciated the attention to an important topic. Based on the reviews, we are likely to accept this manuscript for publication, providing that you modify the manuscript according to the review recommendations. 

It has been really challenging to get 3 reviewers despite inviting 15 reviewers.

Given that the two reviewers highlighted their lack of expertise on the statistical analysis, I offer my comments attached.

Sincerely,

Elvina Viennet, PhD

Deputy Editor

Dear Lance Turtle, 

thank you for your submission. 

It has been really challenging to get 3 reviewers despite inviting 15 reviewers.

Given that the two reviewers highlighted their lack of expertise on the statistical analysis, I offer my comments attached.

Thanks,

Elvina Viennet

Reviewer's Responses to Questions

**Key Review Criteria Required for Acceptance?**

**Methods**

-Are the objectives of the study clearly articulated with a clear testable hypothesis stated?

-Is the study design appropriate to address the stated objectives?

-Is the population clearly described and appropriate for the hypothesis being tested?

-Is the sample size sufficient to ensure adequate power to address the hypothesis being tested?

-Were correct statistical analysis used to support conclusions?

-Are there concerns about ethical or regulatory requirements being met?

Reviewer #1: The plan for data collection, criteria for data inclusion/exclusion, parameters to be analysed from every published work are clearly stated. The methodological details have been extensively described and raw data provided in supplementary material.

Reviewer #2: The objective of the study has been clearly stated and is well substantiated. 

The hypothesis is outlined clearly and the study design is appropriate. Detailed methods in the form of a protocol are presented in Supplementary Data 1.The population is clearly defined, where there is a potential issue with the sample size, these have been discussed. 

I am not particularly qualified to comment on the statistical analysis, however to my best knowledge the authors have used standard methodologies for such studies. 

As this is a meta analysis, I have no particular concerns regarding the ethical and regulatory requirements. The included works are cited as appropriate and the entire data set has been made available.

**Results**

-Does the analysis presented match the analysis plan?

-Are the results clearly and completely presented?

-Are the figures (Tables, Images) of sufficient quality for clarity?

Reviewer #1: There is stepwise description of results, aiming at decreasing heterogeneity in observations and hence arriving at most common factors which are likely to be important for consideration of mice as experimental animals for JE infection. This is done systematically. However there are following issues which could be noted and revised:

1. In general, there are titles for the figures and no legends. In some cases there is not enough description of results in the text leaving figures incomprehensible. This may be amended.

2. More specifically, in Figure 9, forest plot, what are the rhomboids in gray colour depicting?

3. Figure 2 details the parameters used for scoring papers into adhering to norms and standards. However, there are low risk and high risk bias categories shown in each parameter. The reason for defining low vs. high risk is not clear. Is simple absence of documentation in papers, such as 'formal sample size calculation', a criterion for division?

4. What is meant by 'experimental temperature was controlled'? [Fig 2] Does it mean mouse housing temperature? Does it mean mice were treated with anti-pyretics?

5. Can fitted lines/curves be shown for Fig 5b and Fig 6 to improve visual appreciation?

6. Some of the parameters mentioned in Table 2, are more appropriate for human work, and less for mouse work; e.g. treatment and outcome blinded to analysing investigator. They do not seem to be relevant in the final analysis. Can justification for their inclusion in the first place be provided?

7. The authors also document that no paper included in this metanalysis undertook formal sample size calculation. While in principle this is correct, the conflict between statistical requirements for robust data and IACUC imposed restrictions are one major reason for this deficiency observed. This, if acceptable to the authors as a reason, may be elaborated in the text.

Reviewer #2: The analysis presented in consistent with the analysis plan. The results appear to be completely presented, however, they are not clear. 

Figure legends do not sufficiently describe the figures

**Conclusions**

-Are the conclusions supported by the data presented?

-Are the limitations of analysis clearly described?

-Do the authors discuss how these data can be helpful to advance our understanding of the topic under study?

-Is public health relevance addressed?

Reviewer #1: Refinement in analysis has been achieved by eliminating parameters or by eliminating outliers etc. and it makes sense for providing relatively robust guidelines for future use of mice as a model system for JE infection. The final figure, Figure 9, provides a more reliable analysis for future use where mouse strain, mouse age, route of administration, virus dose and virus strain are identified as parameters on which the outcome depends.

Reviewer #2: The conclusions are well supported by the data presented in the study. The limitations of the analysis are discussed, but alternative explanations or expansion of the hypothesis to encompass other factors - such as end point of the study has not been discussed in sufficient detail. The authors make very clear recommendations for future studies - both in the experimental plan as well as reporting of results. This is important and is likely to impact the experimental approach to Japanese Encephalitis Virus pathogenesis in mouse models. The authors have positioned this study within the public health context and have clearly presented the relevance of the study.

**Editorial and Data Presentation Modifications?**

Reviewer #1: Minor explanations and modifications are needed which are listed above under Results section.

Reviewer #2: Perhaps it would be good to present S2 data as a table within the main manuscript?

Figure 1: PRISMA workflow, in the screening stage the numbers do not add up

Figure 2 : Unclear what low risk of bias and high risk of bias are x axis is not labelled

Figure 3 and 4 – while the bar plots of the numbers are useful for comparing the figures, perhaps stating the sample size below the box plots on the x axis and combining a swarm plot may help improve the clarity? 

Figure 9: unclear what is being plotted here in the forest plot. In the text where Figure 9 is referenced, it is unclear what about the figure is being referenced. 

Line 371- where are how is the data available? - This seems to be S9 but it is not cited in the text and the caption for S9 is missing 

Line 385- Perhaps the authors can clarify what is meant by unnecessary waste in this case

Line 391- perhaps the authors can clarify what 'experiments of the kind' means here?

Lines 408-411-This reasoning suggests similarities in blood brain barrier development in mice and humans, but is this the only explanation for the contribution of mice age to heterogeneity? if yes, then such an explanation needs more support with data. 

Line 433- Minor point -Japanese encephalitis virus is not universally a Hazard group 3 pathogen

Figure S4- How was the phylogenetic tree made? How many sequences were included and excluded? How was genotype assigned?

**Summary and General Comments**

Reviewer #1: Bharucha et al undertook extensive review of peer reviewed publications reporting mice as experimental animals for understanding Japanese Virus infection. This is, to the best of my knowledge, and as claimed by the authors, first such attempt to collate, review, analyse data on JEV mortality in mice. Publications from 1935 to 2020 are included. This is a commendable effort, conducted meticulously and is likely to be useful for further research since mice are still very commonly used animals to understand JE pathogenesis, for developing/testing drugs and vaccines. Authors encountered enormous diversity in published data in each of the parameters identified making their task of providing useful information for future work difficult. However, by decreasing heterogeneity within parameters and eliminating some parameters completely from the final analysis they could arrive at a reasonably sound and useful information, which is useful for future researchers.

Work is extremely useful for the scientists, technologists, drug and vaccine developers for JE, since mouse experiments are still needed for drug and vaccine development efforts.

Some of the minor concerns are already highlighted. It will make the review more easily understandable to researchers who may not have extensive understanding of the statistical methods used.

Reviewer #2: Thank you for the opportunity to the review the manuscript titled “Mouse models of Japanese encephalitis virus infection: a systematic review and meta-analysis using a meta-regression approach” by Bharucha et al. The authors have analyzed 127 separate studies which report mortality after inoculation of mice (total of 5026 animals) with virulent strains of Japanese encephalitis virus. They have performed meta regression analyses to identify factors that could account for the heterogeneity observed in the outcome (i.e. mortality). The authors hypothesized and subsequently showed that variations in - virus strain, dose and route of administration,mouse strain, age and sex contribute to observed heterogeneity in the reported mortality. Based on this they constructed a meta regression model with substantially reduced heterogeneity. This work clearly shows the value of such analysis to inform future experiments. In particular, for Japanese encephalitis virus infection of mice, this study provides the framework for better experimental design and suggests reporting standards 

This work is important and the study is clear, the data are convincing. All my concerns both conceptual and editorial are minor.

I have the following concerns regarding the study- given that none of the studies did particularly well on the quality score, how do the authors justify the use of data from such studies? Would setting a quality threshold impact heterogeneity? If not, then is this a relevant quality score? 

It would be good to have some background on why the specific factors were considered as the key variables in the study - are these kind of difference with respect to age and sex of mice for instance known for JEV? What other factors could be included but were not? For instance end point or duration of experiments is another variable, why was this not included in the study?

One would expect the amount of virus inoculated to correlate strongly with mortality, however, this is not clear from Figure 5. In the more homogeneous final model, does the equivalent of Figure 5 resolve this better?

PLOS authors have the option to publish the peer review history of their article (what does this mean?). If published, this will include your full peer review and any attached files.

Reviewer #1: No

Reviewer #2: No

Figure Files:

Data Requirements:

Reproducibility:

References

---

## [Decision Letter · Decision Letter 1]

19 Dec 2021

Dear Dr Turtle,

We are pleased to inform you that your manuscript 'Mouse models of Japanese encephalitis virus infection: a systematic review and meta-analysis using a meta-regression approach' has been provisionally accepted for publication in PLOS Neglected Tropical Diseases.

Best regards,

Elvina Viennet, PhD

Deputy Editor

Elvina Viennet

Deputy Editor

Reviewer's Responses to Questions

**Key Review Criteria Required for Acceptance?**

**Methods**

-Are the objectives of the study clearly articulated with a clear testable hypothesis stated?

-Is the study design appropriate to address the stated objectives?

-Is the population clearly described and appropriate for the hypothesis being tested?

-Is the sample size sufficient to ensure adequate power to address the hypothesis being tested?

-Were correct statistical analysis used to support conclusions?

-Are there concerns about ethical or regulatory requirements being met?

Reviewer #1: (No Response)

Reviewer #2: (No Response)

**Results**

-Does the analysis presented match the analysis plan?

-Are the results clearly and completely presented?

-Are the figures (Tables, Images) of sufficient quality for clarity?

Reviewer #1: (No Response)

Reviewer #2: (No Response)

**Conclusions**

-Are the conclusions supported by the data presented?

-Are the limitations of analysis clearly described?

-Do the authors discuss how these data can be helpful to advance our understanding of the topic under study?

-Is public health relevance addressed?

Reviewer #1: (No Response)

Reviewer #2: (No Response)

**Editorial and Data Presentation Modifications?**

Reviewer #1: (No Response)

Reviewer #2: (No Response)

**Summary and General Comments**

Reviewer #1: The manuscript has been revised as suggested and may be accepted.

Two minor proofing suggestions:

line 281. Instead of Fig 3C, it should be 3D.

A few typos got introduced with revision of the manuscript!

Reviewer #2: (No Response)

PLOS authors have the option to publish the peer review history of their article (what does this mean?). If published, this will include your full peer review and any attached files.

Reviewer #1: No

Reviewer #2: No

---

## [Editor Report · Acceptance letter]

7 Feb 2022

Dear Dr Turtle,

We are delighted to inform you that your manuscript, "Mouse models of Japanese encephalitis virus infection: a systematic review and meta-analysis using a meta-regression approach," has been formally accepted for publication in PLOS Neglected Tropical Diseases.

Best regards,

Shaden Kamhawi

co-Editor-in-Chief

Paul Brindley

co-Editor-in-Chief
